# Turn-by-Turn Driving Navigation: Leveraging Sequence Model for Real-time Audio Instructions

## Abstract

Turn-by-turn (TBT) navigation systems are integral to modern driving experiences, providing real-time audio instructions to guide drivers safely to destinations. However, existing audio instruction policy often rely on rule-based approaches that struggle to balance informational content with cognitive load, potentially leading to driver confusion or missed turns in complex environments. To overcome these difficulties, we first model the generation of audio instructions as a multi-task learning problem by decomposing the audio content into combinations of modular elements. Then, we propose a novel deep learning framework that leverages the powerful spatiotemporal information processing capabilities of Transformers and the strong multi-task learning abilities of Mixture of Experts (MoE) to generate real-time, context-aware audio instructions for TBT driving navigation. A cloud-edge collaborative architecture is implemented to handle the computational demands of the model, ensuring scalability and real-time performance for practical applications. Experimental results in the real world demonstrate that the proposed method significantly reduces the yaw rate compared to traditional methods, delivering clearer and more effective audio instructions. This is the first large-scale application of deep learning in driving audio navigation, marking a substantial advancement in intelligent transportation and driving assistance technologies.

## 1 Introduction

Navigation system in the era of mobile internet have improved the driving experience by providing drivers with route information and directions in real-time via visual and audio instruction on the navigation terminal Yang et al. (2024). Compared with visual information, drivers tend to rely more on audio instructions for the sake of driving safety Zhong et al. (2022). However, current turn-by-turn (TBT) driving navigation Fabrikant (2023) often find it challenging to strike a balance between yaw rate, play timing, and play density for audio instruction. Overly complex audios can increase the driver's cognitive load, making it difficult to understand information quickly and safely. Conversely, a simplistic audio may fail to provide sufficient guidance for navigating complex intersections and intricate road networks, leading to yaw and compromised safety Large & Burnett (2014).

The principal difficulty in generating real-time audio instructions lies in how to play the accurate and complete audio at the correct time. Current methods typically use rule-based policies or pre-defined configuration tables Jensen et al. (2010); Large & Burnett (2014); Yang et al. (2021) that lack the flexibility, makes it easy to fall into the seesaw effect between yaw rate, play timing, and audio density. These approaches may result in audio instructions that are either too general, failing to convey critical information, or too verbose, overloading the driver with unnecessary details.

To address these challenges, we propose a novel pipeline for audio instruction generation: firstly, decomposing the audio context into modular elements, then the audio instruction model in charge of element recall, play timing, and order selection, and finally generating coherent speech via text-to-speech (TTS) module Kaur & Singh (2023). This pipeline transforms the complex task of audio instruction generation into manageable components, allowing for precise and adaptable instructions that effectively balance informational content with cognitive load.

Building upon this formalization, we introduce the first deep learning framework utilizing sequence models for real-time, context-aware audio instruction generation in practical TBT driving navigation. Our method captures the complex dependencies and variations inherent in driving scenarios. By leveraging sequence modeling, our approach effectively handles complex intersections and effectively reduces the yaw rate.

Our main contributions are as follows:

- **The First Deep Learning Based TBT Driving Navigation:** To the best of our knowledge, we are the first to implement the deep learning based audio instruction for practical applications, utilizing sequence models to capture spatiotemporal dependencies and address the seesaw effect between yaw rate, play timing, and audio density.

- **Novel Audio Navigation Framework:** To the best of our knowledge, we are the first to formulate audio instructions as elements recall, play trigger, and order prediction, which helps probe multi-task learning quantitatively and opens a new paradigm for TBT driving navigation research and application.

- **Data-Driven Paradigm for TBT Optimization:** We introduce the data-driven paradigm for optimizing TBT driving navigation, which shifts from rule-based to data-driven optimization results in continuous performance improvements.

## 2 PRELIMINARIES AND BACKGROUND

TBT driving navigation refers to a navigational aid system that provides step-by-step instructions to drivers, guiding them from a starting location to the destination. This system utilizes real-time driving data, often incorporating Global Positioning System (GPS) technology, to help with wayfinding problems during driving Schwering et al. (2017); Fabrikant (2023). Instructions are typically delivered via audio prompts and visual cues, indicating when and where to make turns, lane changes, and other operations.

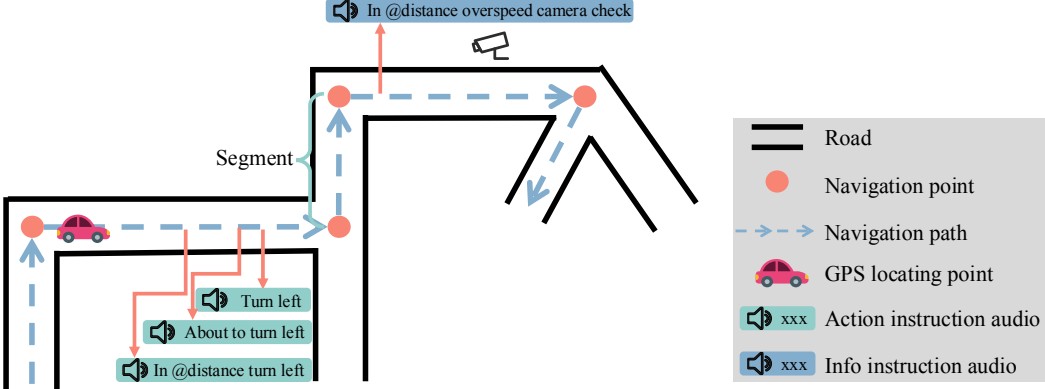

Figure 1: **TBT driving navigation.** The figure displays the essential cue content and key terminology used in the TBT driving navigation, along with the paradigm of the audio instruction while the car moves through the path.

The basic flow of audio instructions is presented in Figure 1, illustrating key concepts within the TBT driving navigation we have defined. The navigation point represents the location where the navigation system expects the driver to make a steering to avoid yawing, typically at a fork in the road. The directed connecting line between two neighboring navigation points is called as segment. All segments form the navigation path. The primary goal of the audio instruction policy during navigation is to select the appropriate content at the proper time to prevent the driver from yawing or violating traffic rules, where yawing refers to the driver deviating from the planned route provided by the navigation system, which usually means that the instruction content is wrong or poorly timed, causing the driver to go the wrong way.

The audio content must follow a standardized approach and be brief to ensure quick comprehension by the driver. Consequently, when generating instruction audio, sentence diversity is not a consideration, unlike other language generation tasks. This allows us to organize instructional audio content using key elements and consistent connectors, where the key information entities related to driving are called elements. We define a set of elements $\mathcal{E}_{\text{elem}} = \{\mathcal{E}_{\text{action}}, \mathcal{E}_{\text{info}}\}$ that encapsulate the key information to be conveyed through audio instructions. These elements are categorized into:

- **Action Elements** $\mathcal{E}_{\text{action}}$: Elements that require the driver needs to turn the wheel following the audio instruction, such as "turn left" or "merge right."

- **Info Elements** $\mathcal{E}_{\text{info}}$: Elements that provide information without requiring to turn the wheel, such as "speed camera ahead" or "exceeding speed limit, reduce your speed".

Each audio instruction contains multiple action elements, at most one info element, play timing, and play order in the segment. We have listed all the element types in the appendix A.1. Based on this, we structure an instruction audio as: the elements that need to be revealed, the order within the segment, and the play timing of the audio. The play timing is indicated by the relative distance from the audio playing position to the navigation point. The term "order" refers to the position of the current audio in the segment after all audios are sorted by play position. It is related to the selection of the connectors. Figure 1 shows the connections of play order in the segment and the organization of audio content through an example: 3 green boxes represent 3 different instruction audios, each with the same element "turn left". However, the connectors for the element vary depending on the play order, resulting in corresponding content changes.

Many studies have been devoted to optimizing the content and timing of audio navigation messages to improve driving safety and experience. Some researchers Yang et al. (2021); Bian et al. (2021) investigated the effects of different cue timing patterns and cue message types through a driving simulation experiment. They found that the interaction between cue timing and cue messages significantly affected drivers' psychological state and vehicle operation. And some studies investigated the initiation function of in-vehicle audio commands and found that audio commands can effectively facilitate drivers' quick and safe responses to the road environment Keyes et al. (2019). Wunderlich et al. propose to use landmark augmented audio navigation to enhance the spatial awareness for drivers Wunderlich et al. (2023).

While these studies have advanced the understanding of how navigation prompt messages impact driver behavior, they primarily rely on handcraft or rule-based audio instructions. While these methods have yielded commendable results in simulation or specific scenes, the ever-changing and intricate nature of real-world roadways renders it manifestly inadequate to depend solely on handcrafted or rule-based audio instructions to guarantee effective instructions under all conditions. Enhancing the audio density might reduce yaw rates at straightforward intersections; however, the consequent increase in instructional content can infringe upon the timing of subsequent audio at complex intersections. This encroachment can lead to the compression or omission of critical elements, thereby inducing driver confusion and route deviation. This dilemma is referred to as the seesaw effect among yaw rate, audio density, and play timing in audio instruction. Nonetheless, by integrating deep learning into TBT driving navigation and training neural network-based audio instruction policy on carefully curated high-quality data that exemplifies effective guidance, it becomes feasible to leverage the robust generalization capabilities of neural networks. Such an approach holds the potential to transcend the seesaw effect mentioned above, thereby further enhancing the performance and reliability of dio instructionsaui systems.

## 3 METHOD

In this section, we will delve into the details of our approach, covering the following aspects: Firstly, Section 3.1 formalizes the audio instruction problem as multi-task learning. Then, Section 3.2 introduces the sequence model for audio instruction, along with the cloud-edge collaboration for model deployment. Finally, Section 3.3 outlines the model training.

## 3.1 PROBLEM FORMALIZATION

To address the challenges in generating real-time, context-aware instructions, we model the audio instruction in TBT driving navigation as a multi-task learning problem. Enables the model to optimize the necessary components for generating the audio simultaneously.

As presented in Section 2, the audios within each segment have a strong spatiotemporal correlation, so segments are selected as the granularity for modeling driving scenarios oriented towards audio instruction. We sample features for the audio instruction model in segments at 1-second intervals: $x_{t_1}, x_{t_2}, \ldots, x_{t_T}$, where $T$ is the total number of time steps the car passes the segment. The feature components are listed in Appendix Table 9.

Considering the importance of feature sequences for the audio instruction task, we aim to learn a function that maps the input sequence data to several outputs to compose audio instructions:

$$f : \mathcal{X}_t \to \{y_{\text{trigger}}, y_{\text{action}}, y_{\text{info}}, y_{\text{vo}}\}, \tag{1}$$

where $\mathcal{X}_t = \{x_{t-n}, ..., x_{t-1}, x_t\}$ represents the input sequence features, where $n$ is the sequence length. $y_{\text{trigger}}$ represents the ratio of the relative distance from the audio play position to the navigation point $d_{\text{pp}}$ and the relative distance from the current driver's position to the navigation point $d_{\text{np}}$: $y_{\text{trigger}} = \frac{d_{\text{pp}}}{d_{\text{np}}}$, indicating the audio play timing. $y_{\text{action}} \in \{0,1\}^{|\mathcal{E}_{\text{action}}|}$ is a binary vector indicating which action elements should be included in the audio. $y_{\text{info}} \in \{0,1\}^{|\mathcal{E}_{\text{info}}|}$ is a binary vector indicating which info elements should be included. $y_{\text{vo}} \in \{1, 2, \ldots, O\}$ represents the play order of the audio instruction within the segment, where $O$ is the maximum number of possible orders.

By formalizing the problem in this way, we can model the mapping function from input features to outputs via deep neural networks based on maximum likelihood estimation to fit the distribution of high-quality data to learn a better audio instruction policy.

## 3.2 SEQUENCE MODELING

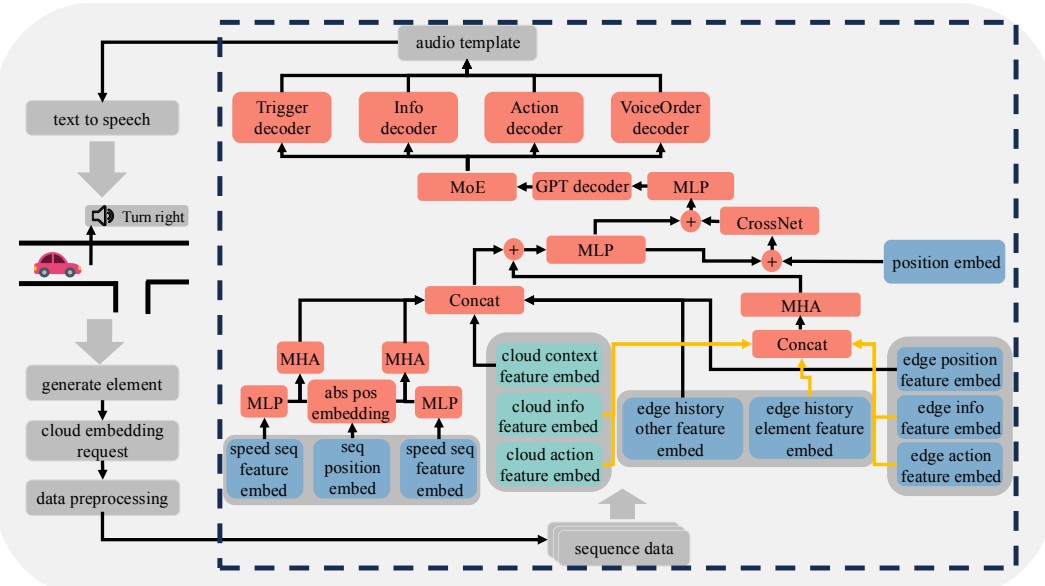

Figure 2: **Overview of TBT audio instruction model.** In the dashed box is the proposed audio instruction model. The red box represents the model component, the blue box represents the edge-side data embedding, and the green box represents the cloud-side data embedding. To the left of the dashed box is the engineering design for model real-time playing and data preparation.

Figure 2 shows the audio instruction model and its application framework in the TBT driving navigation system. The model adopts a cloud-edge cooperative architecture, considering scalability,

real-time computation, and resource optimization. The responsibility of the cloud side is to embed features that are relatively static in the segment. The edge side is responsible for embedding other features that have high real-time requirements and then performing model inference on the edge side along with the feature embedding sent down from the cloud. Detailed description and analysis of the advantages of the cloud-edge architecture is provided in the appendix A.2.

Since the current audio content generation is strongly correlated with the historical play within the segment we combine the features of the current timestep with the features of the previous $n-1$ moments in chronological order as sequence data $\mathcal{X}_t = \{x_{t-n}, ..., x_{t-1}, x_t\}$ for model input. If there is less than $n$ audio data in the segment, zero padding is applied to the missing portion of the sequence. Unlike conventional sequence labeling tasks, the TBT audio instruction task encounters challenges in calculating sequence loss Huang et al. (2015) due to the inability to predetermine the input sequence. Furthermore, the TBT audio instruction task does not conform to the autoregressive training paradigm employed by large language models Zhao et al. (2023), as the current audio instruction cannot be inferred solely from historical audios but is also strongly correlated with the current driving context. Therefore, we have tailored a sequence model for TBT audio instruction by integrating the spatiotemporal information processing capabilities of the Transformer.

Upon completion of model inference, the predicted timing by the trigger head is used to assemble a complete sentence by combining the inferred action via the action head and information elements via the info head with connecting word templates based on the voice order head. The sentence is then converted into speech via a Text-to-Speech (TTS) module and ultimately plays in the driver's navigation terminal. This process is iteratively executed throughout the entire navigation path, constituting a comprehensive TBT audio instruction system.

It should be noted that the action head and the info head are responsible for recalling elements required in the current play content, while all candidate elements are given to the model as input features. The candidate elements in the input features are generated per segment by the scheduling unit based on the current road graph and path planning information.

The model architecture illustrated within the dashed box in Figure 2 can be broadly divided into 4 levels: the *Feature Encoder*, the *Deep CrossNet* Zheng et al. (2018), the *GPT Decoder* Brown et al. (2020), and the *MoE Prediction Layer*:

Initially, the sequence data undergoes *Feature Encoder*. All element-related features are embedded and concatenated, which are indicated by the orange arrows in Figure 2. The element embeddings are then fully encoded and mixed through the Multi-Head Attention (MHA). After that, the mixed element embedding is encoded via a Multi-Layer Perceptron (MLP) along with other input features. The *Feature Encoder* transforms the high-dimensional sparse feature representations into low-dimensional dense vectors while capturing and preserving the intrinsic structure and semantic information of the data, facilitating subsequent model processing.

Subsequently, the encoded element features are combined with position embeddings. Different from the existing position embedding method Vaswani et al. (2017); Devlin et al. (2019); Su et al. (2024), we integrate domain-specific prior knowledge to transform the conventional absolute position embedding into a combination of temporal sequence encoding and spatial semantic encoding. Temporal sequence encoding targets each effective time slice in the sequence, performing reverse indexing and learning through a position embedding matrix. As for spatial semantic encoding, considering that the density of audio instruction increases as the car approaches the next navigation point $d_{\mathrm{np}}$, a distance-based weight discount $\gamma$ is applied. The greater $d_{\mathrm{np}}$, the smaller $\gamma$. A detailed description of the position embedding is given in Appendix A.3.

The data is then processed through the *Deep CrossNet*, which constructs and learns high-order cross-feature combinations. The data is then merged through a residual connection and further encoded by an MLP before being input into the next part.

The *GPT Decoder* is designed to exploit the spatiotemporal coupling information inherent in sequential data. Each time slice in the data can establish associations with other time slices in the sequence, rather than relying solely on adjacent time slice data. By computing the attention map, the model adaptively captures the rich semantic information within the sequence. We choose a GPT-like architecture for the spatiotemporal data processing module because its self-supervised training paradigm naturally aligns with predicting current audio based on sequence features. This

autoregressive framework allows the model to effectively learn temporal dependencies within the sequence data, enhancing its capacity to generate accurate and context-aware audio instructions.

Finally, the sequence data processed by the *GPT Decoder* is fed into the *MoE Prediction Layer* for multi-task learning. This layer simultaneously learns to predict 4 sub-tasks necessary for generating an instruction audio: the audio trigger time, the action-type elements included in the audio, the information-type elements included in the audio, and the audio play order within the segment. The underlying shared features are learned using the MoE. Through different combinations of these expert networks, each subsequent sub-task head can efficiently focus on the features most pertinent to its specific requirements.

### 3.3 MODEL TRAINING

As illustrated in Figure 2, the outputs of the TBT audio instruction model are divided into 4 sub-task outputs: trigger, action, info, and voice order. Each sub-task has a unique loss function tailored to its specific prediction task:

The trigger decoder is responsible for predicting the normalized play timing of the audio instruction, with its scalar output $\hat{y}_{\text{trigger}} \in [0, 1]$. The mean squared error (MSE) is employed as the loss function:

$$\mathcal{L}_{\text{trigger}} = (y_{\text{trigger}} - \hat{y}_{\text{trigger}})^2, \tag{2}$$

where $y_{\text{trigger}}$ is the audio play timing label.

The action decoder is responsible for predicting the action-type elements that should be included in the audio instruction from the available action elements in the segment. Since a single audio instruction may contain multiple action-type elements, the action decoder outputs a prediction vector with a length equal to the number of action-type elements. Each probability prediction value $p_i \in [0, 1]$ in the prediction vector corresponds to an action element $i$. If $p_i > 0.5$, the action element is included in the audio; otherwise, it is excluded. The loss function is defined as follows:

$$\mathcal{L}_{\text{action}} = (\boldsymbol{y}_{\text{action}} - \hat{\boldsymbol{y}}_{\text{action}})^2, \tag{3}$$

where $\boldsymbol{y}_{\text{action}}$ is the label vector for action elements. The position corresponding to the element contained in the audio is 1, otherwise 0.

The info decoder is tasked with predicting the information-type elements that should be included in the audio instruction from the available information elements in the segment. Since one audio instruction can contain at most one information-type element, cross-entropy loss is employed:

$$\mathcal{L}_{\text{info}} = -\sum_{i=1}^{25} \boldsymbol{y}_{\text{info},i} \cdot \log \hat{\boldsymbol{y}}_{\text{info},i}, \tag{4}$$

where 25 is the total number of information-type elements plus one, with the additional position indicating the probability that no information-type element is included in the audio instruction. $\boldsymbol{y}_{\text{info}}$ is the one-hot label representing the information-type element included in the audio instruction.

The voice order decoder is responsible for predicting the play order of the audio instruction within the segment. It uses one-hot encoding to classify the order into five categories, ranging from 0 to 4, where 0 indicates that the current audio instruction should not be played, and 1 to 4 represents the play order of the audio relative to the endpoint of the navigation segment. Cross-entropy loss is employed:

$$\mathcal{L}_{\text{vo}} = -\sum_{i=1}^{5} \boldsymbol{y}_{\text{vo},i} \cdot \log \hat{\boldsymbol{y}}_{\text{vo},i}, \tag{5}$$

where $\boldsymbol{y}_{\text{vo}}$ is a one-hot encoded label indicating the play order of the audio instruction within the segment. $\hat{\boldsymbol{y}}_{\text{vo}}$ represents the predicted play order of the current audio instruction within the segment.

To address the issue of varying learning difficulties across different sub-tasks in multi-task training and to avoid subpar performance in certain sub-tasks, the total loss function is computed using the geometric mean:

$$\mathcal{L}_{\text{total}} = (\mathcal{L}_{\text{trigger}} \cdot \mathcal{L}_{\text{action}} \cdot \mathcal{L}_{\text{info}} \cdot \mathcal{L}_{\text{vo}})^{\frac{1}{4}}, \tag{6}$$

where $\mathcal{L}_{\text{trigger}}$, $\mathcal{L}_{\text{action}}$, $\mathcal{L}_{\text{info}}$, and $\mathcal{L}_{\text{vo}}$ are the individual loss functions for the trigger, action, info, and voice order decoders, respectively. The geometric mean ensures a balanced contribution from each sub-task, mitigating the risk of any single sub-task dominating the overall training process and leading to more robust model performance across all tasks.

## 4 EXPERIMENTS

In Section 4.1, we first introduce the dataset and model configurations. Subsequently, in Section 4.2, we demonstrate the advantages of our approach through an AB test by deploying the model in real-world driving navigation and comparing it with the existing HMM-based TBT audio instruction policy. Then, in Section 4.3, we evaluate the impact of key components of the model on the overall performance of the neural network through offline ablation experiments. Finally, in Section 4.4, we randomly invited 100 drivers to participate in a blind evaluation of our model and the existing HMM-based TBT audio instruction policy. This evaluation covered 6 scenarios that are prone to yaw. The purpose of this assessment was to focus on the in-car experience of drivers in order to evaluate the effectiveness of our method on another dimension.

### 4.1 DATASET AND MODEL CONFIGURATIONS

To train and evaluate our TBT audio instruction model, we construct a large-scale dataset derived from real-world driving navigation logs. We collect navigation trajectory data from actual drivers over 8 days, from June 11 to June 18, 2023. The audio instruction policy for online data collection is a language generation policy modeled using Hidden Markov Models, which will be denoted as "HMM" in subsequent experiments. HMM is described in more detail in Appendix A.4. To maintain data quality and relevance, we have strict selection criteria. Only navigation paths initiated by cars are included, and we exclude trajectories with muted driver terminals, abnormal driving speeds, yawing, and GPS drifting during navigation. We then apply a secondary filter to the navigation trajectories based on our domain knowledge. This helps us to identify high-quality navigation trajectories with normal element transmission and audio play timing that meets the expectations within each segment. Finally, these filtered high-quality real online trajectories are used to construct the dataset for the audio instruction model training. A more detailed dataset generation process is presented in Appendix A.4.

The datasets are partitioned into training, validation, and test sets. Feature standardization is performed using the mean and standard deviation calculated over the dataset. Sequential sample data are constructed by concatenating individual positioning point samples. The final dataset comprises approximately 1.56 billion sequence samples, with 1.1 billion samples in the training set (including 10 million supreme quality samples for supervised fine-tuning), 140 million samples in the validation set, and 320 million samples in the test set. This extensive dataset provides a robust foundation for training model and assessing its performance in generating effective TBT audio instructions.

The model input features have 2139 dimensions, as detailed in Table 9 in the appendix. The length of sequence data is set to 3. The MoE part contain 3 experts. The model comprises 4 output heads: the trigger head outputs a scalar activated by the sigmoid function, the action head outputs a 28-dimensional vector also activated by the sigmoid function, the info head outputs a 25-dimensional vector activated by the softmax function, and the voice order head outputs a 5-dimensional vector activated by the softmax function. The model parameters are 1,147,943. During training, the learning rate is set to 0.001, the batch size is 800, and the model is trained for 400,000 steps, which takes approximately 38 hours on 8 NVIDIA T4 GPUs. Additional training parameters are provided in Table 10 in the appendix.

For online deployment, the model trained with float32 is converted to the float16 model. The model is split into two parts: one for the edge device and one for the cloud server, which is deployed on the user's navigation terminal and cloud server, respectively. The cloud part primarily consists of cloud feature embeddings. After converting the model to ONNX using TensorRT, the model size is approximately 807 KB, and it is inferred on the cloud server. The end part of the model is first converted to ONNX using TensorRT and then to MNN Jiang et al. (2020), resulting in a model size of approximately 2.3 MB. This part of the model is inferred on the user's navigation terminal.

## 4.2 REAL-WORLD A/B TEST

To empirically validate the effectiveness of our model, we deploy the TBT audio instruction model online, in order to compare with the HMM audio instruction policy widely used in driving navigation system. Our navigation system offers 4 modes to meet the different needs of drivers: detail, concise, minimalist, and intelligent. Drivers can select these modes based on their preferences. The detail and intelligent modes have more frequent audio prompts and are suitable for navigating unfamiliar roads, while the concise and minimalist modes have fewer prompts and are ideal for familiar routes.

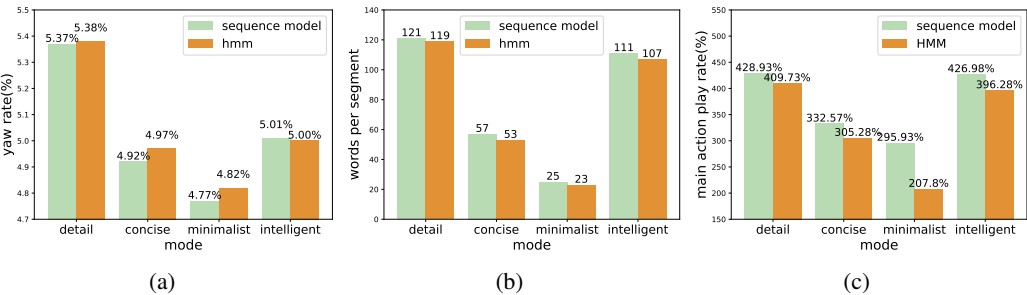

(a)                                  (b)                                  (c)

Figure 3: **Real-world A/B test results.** Green represents the sequence model and orange represents the existing HMM policy. (a) Yaw rate, lower yaw rates represent more effective audio instruction; (b) Words density, lower word density means more streamlined audio; (c) Main action element play rate, higher means more frequent TBT message alerts.

The A/B test experiment period collected vehicle navigation data via our navigation system from August 28, 2024, to September 3, 2024, spanning a week and encompassing data from about 600 million segments. The primary results of the online experiment are illustrated in Figure 3. The comparison mainly focuses on the yaw rate, the average words played per segment and the average play density of elements. The yaw rate is the ratio of yaw segments to all segments. Lower yaw rate means more accurate audio instruction. The play density of an element is define as the element play counts divided by the number of segments which can play this element. The average word count is positively related to the driver's difficulty in comprehending the content of the instruction audio, and the element play rate is positively related to the amount of information in the output content of the audio instruction model. In general, the more elements that are played with fewer average words represents a better content organization ability of the audio instruction model.

Compared to existing HMM-based approaches, our sequence model achieves a significant reduction in yaw rate. As shown in Figure 3 (a), except for the intelligent mode which has a slightly higher yaw, our model achieves a significant yaw rate reduction on all other modes, especially concise and minimalist. Note that since our daily online user volume is billions, $0.01\%$ reduction in the yaw rate represents success in helping hundreds of thousands people drive to their destinations correctly. So the yaw improvement on the order of $0.01\%$ is also significant.

Moreover, the increase in the average number of words played per segment, as illustrated in Figures 3(b) and 3(c), indicates that our model incorporates more main action elements with only 2-4 words increase. More element play rates are revealed in Appendix A.5, and for most of them, the play rate increases. This indicates that our sequence model breaks through the seesaw effect of yaw rate, play density, and timing: with almost no increase in audio play words, the audio information density increase is achieved by significantly increasing the element play rates, and the impact on the play timing is few because the audio text length is almost unchanged. This in turn reduces the yaw rate.

In summary, the results of the real-world A/B test validate the effectiveness of our method in providing real-time, context-aware audio instructions that significantly reduce the yaw rate. In addition, our model can adapt to drivers' diverse navigation detail preferences, as evidenced by its superior performance in different modes, highlighting its robustness and generalizability under real-world driving conditions.

## 4.3 ABLATION STUDY

To investigate the necessity and effectiveness of each component in our model, we conducted a comprehensive ablation study offline by systematically removing or altering individual components and observing the impact on overall performance. The results are summarized in Table 1.

| Model Type | Trigger 10m | Trigger 30m | Action | Info | VoiceOrder |
|---|---|---|---|---|---|
| our model | 83.3% | 96.3% | 97.0% | 98.6% | 90.7% |
| BERT decoder | −2.0% | −0.8% | −0.1% | −0.3% | −0.1% |
| w/o position embedding | −2.5% | −2.1% | −0.2% | −0.1% | −1.9% |
| w/o MoE | −0.5% | −0.1% | −0.4% | −0.2% | −0.5% |
| w/o CrossNet | −3.0% | −3.8% | −4.1% | −2.0% | −2.1% |
| w/o sequence | −5.5% | −1.4% | −0.3% | −0.2% | −1.5% |

Table 1: Ablation study

The metrics used in this section are defined as follows: **Trigger 10m**: The accuracy of the trigger head predictions within a 10-meter range; **Trigger 30m**: The accuracy of the trigger head predictions within a 30-meter range; **Action**: The accuracy of the action head predictions; **Info**: The accuracy of the info head predictions; **VoiceOrder**: The accuracy of the voice order head predictions.

Our model achieved a trigger 10m accuracy of $83.3\%$, a trigger 30m accuracy of $96.3\%$, an action accuracy of $97.0\%$, an info accuracy of $98.6\%$, and a voice order accuracy of $90.7\%$. These results affirm the robustness and high performance of our proposed method.

When we replaced the GPT decoder with a BERT decoder, we observed a slight decrease in performance across all metrics. Specifically, trigger 10m and trigger 30m accuracies dropped by $2.0\%$ and $0.8\%$, respectively, while action, info, and voice order accuracies decreased marginally. This indicates that the GPT decoder is better suited for capturing the spatial and temporal dependencies of sequence data in audio instruction tasks compared to BERT.

Removing the position embedding resulted in a more pronounced decline in performance, particularly for trigger 10m, trigger 30m, and voice order accuracies. This indicates that sequence and element position information is critical to the model prediction of timing and order.

Eliminating the MoE component led to a moderate reduction in performance, with the most significant impact observed on the action and voice order accuracies. This suggests that the multi-task learning capabilities of the MoE framework play a crucial role in effectively handling the diverse and interrelated sub-tasks of the TBT audio instruction model.

The removal of the CrossNet resulted in the most substantial performance degradation across all metrics, with Trigger 10m and Action accuracies decreasing by $3.0\%$ and $4.1\%$, respectively. This highlights the critical role of high-order feature interactions in capturing the complex relationships between different input features.

Finally, when we omitted the sequential input features, which means reducing the sequence length from 3 to 1, we observed a significant drop in Trigger 10m accuracy by $5.5\%$ and a noticeable decline in other metrics. This demonstrates the necessity of incorporating sequence information for providing accurate and context-aware audio instructions. More experiments on sequence length and selection of the number of MoE experts are presented in Appendix A.6.

In conclusion, the ablation study confirms that each component of our model contributes significantly to its overall performance. The superior results achieved by our full model validate the design choices made during development and underscore the effectiveness of leveraging advanced deep learning techniques for real-time, context-aware TBT audio instructions.

## 4.4 BLIND EVALUATION

To further assess the effectiveness of our proposed sequence model in real-world driving scenarios, we conducted a blind evaluation comparing our model's TBT audio instructions with those generated by the existing HMM-based policy. The goal was to evaluate the in-car experience from

the driver's perspective, focusing on how well the audio instructions aid in navigating challenging driving situations that are prone to yaw.

We randomly recruited 100 drivers to participate in this study. Each driver was presented with pairs of audio instructions generated by our model and the HMM-based policy for six different driving scenarios known to cause navigational difficulties, which are described in detail in Appendix A.7.

For each scenario, the drivers were asked to listen to the audio instructions without knowing which model generated them and to rate which one they preferred or if they found them equally effective. The results are summarized in Table 2.

| Scenario | Sequence Model Better | HMM Better | No Difference |
|---|---|---|---|
| Near double bend | 21% | 12% | 67% |
| Mix fork | 28% | 22% | 50% |
| Roundabout | 54% | 18% | 28% |
| Short segment | 45% | 15% | 40% |
| Double traffic light | 21% | 17% | 62% |
| Tunnel | 11% | 18% | 69% |

Table 2: Results of the blind evaluation comparing the Sequence Model and HMM policy across different challenging driving scenarios.

In the *roundabout* and *short segment*, a significant proportion of drivers preferred the audio instructions generated by our sequence model over those from the HMM-based policy. These results indicate that our model provides clearer and more effective guidance in complex scenarios where precise timing and content of instructions are critical. The enhanced performance in these situations can be attributed to our model's ability to capture spatiotemporal dependencies and generate context-aware audio instructions that adapt to the driving environment in real time.

In the *near double bend*, *mix fork*, and *double traffic light*, the majority of drivers found little difference between the two models, with a slight preference for our sequence model. It suggests that while both models perform adequately in these scenarios, our model still offers marginal improvements.

In the *tunnel* scenario, slightly more drivers preferred the HMM policy (18%) over our model (11%). This may be due to the unique challenges posed by tunnels, such as GPS signal loss, which can affect real-time data processing.

Overall, the blind evaluation demonstrates that our sequence model outperforms the traditional HMM-based policy in delivering timely and contextually appropriate audio instructions, especially in complex driving conditions prone to navigation errors. By effectively balancing informational content with cognitive load and adapting to dynamic driving contexts, our model enhances the driver's situational awareness and decision-making, thereby improving safety and navigation efficiency. These findings validate the practical applicability and advantages of leveraging deep learning in TBT audio instruction systems.

## 5 CONCLUSION

In this paper, we introduce a novel deep-learning framework leveraging sequence models for real-time, context-aware audio instructions in TBT driving navigation. By formalizing the audio instruction generation into modular elements and utilizing a cloud-edge collaborative architecture, our approach effectively balances informational content with cognitive load. Extensive experiments, including real-world A/B tests and blind evaluations, demonstrated that our model significantly reduces yaw rates compared to HMM-based policies, successfully incorporates more informative elements into audio instructions without overwhelming the driver. And the ablation studies confirmed the critical contributions of each component in our model.

Our method represents the first large-scale application of deep learning in practical driving audio navigation, marking a substantial advancement in intelligent transportation technologies. Future work will focus on further optimizing model performance in complex scenarios and exploring personalized navigation experiences by integrating individual driver preferences and behaviors.

## 6 REPRODUCIBILITY STATEMENT

We have made significant efforts to ensure the reproducibility of our results presented in this paper. The complete implementation of our model, including the model structure, inference code, and part of the training code, is provided in the supplementary materials. This includes detailed descriptions of the model architecture, hyperparameter settings, training procedures, and the cloud-edge collaborative deployment as discussed in Section 3 and detailed in Appendix A.2 and Appendix A.8. While our dataset and complete training flow cannot be open-sourced due to user privacy and commercial confidentiality considerations, we have provided a comprehensive explanation of the data collection process, selection criteria, preprocessing steps, and dataset composition in Section 4.1 and Appendix A.4. We have also thoroughly described the feature engineering and input representations in Appendix A.1 and Appendix A.8. All experimental settings, evaluation metrics, and analysis methods are detailed in Section 4, with additional experimental results and ablation studies presented in Section 4.3 and Appendices A.5 and A.6. By providing the code and comprehensive descriptions of our methodologies and experiments, we aim to facilitate the replication and validation of our work by the research community.

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

# A APPENDIX

## A.1 ELEMENT TYPE AND SUB-TYPE

| Element name | Onehot index | Description |
|---|---|---|
| Mainaction | 0 | The main action element, which represents the action that the driver needs to take at the next navigation point, must be present in every segment. |
| Assistaction | 1 | The assist action element. Supplementary information to the main action element usually reveals information about the next segment so that drivers can better understand instruction. |
| Slope | 2 | Existence of uphill or downhill. |
| Lane | 3 | Lane related instructions for multi-lane roads |
| WideLane | 4 | Wide lane reminder |
| MixLane | 5 | Driving lane reminder |
| TunnelLane | 6 | Driving lane reminder for entering tunnel |
| Longsolidlane | 7 | Existence of long solid lines |
| Linkturn | 8 | Significant curvature of the road in the middle of the segment, but no turnoffs |
| Mixfork | 9 | Two same direction turnoffs close to each other at the navigation point($d_{\mathrm{np}} = 0$) |
| AroundFork | 10 | Pass the roundabout exit |
| ExitRoad | 11 | Exit road for highway or urban expressway |
| TunnelSimpleLane | 12 | Tunnel lane confirmation |
| Nextbrname | 13 | Enter XXX road and head towards XXX |
| NextMainaction | 14 | Main action element in next segment |
| NextAssistaction | 15 | Assist action element in next segment |
| NextSlope | 16 | Slope element in next segment |
| NextSegNextbrname | 17 | Nextbrname element in next segment |
| NextLane | 18 | Lane element in next segment |
| NextNextAct | 19 | Main action element in the segment after next |
| NextExitRoad | 20 | ExitRoad element in next segment |
| SolidLane | 21 | Existence of solid lines |
| Nonnavigation | 22 | Fork not at navigation points, generally straight ahead |
| Mixfork1 | 23 | Two same direction turnoffs close to each other before the navigation point($d_{\mathrm{np}} > 0$) |
| NextMixfork0 | 24 | Mixfork0 element in next segment |
| ShortNonNaviLane | 25 | Next sub-segment lane |
| RTKSingelPlay | 26 | Real-time kinematic lane instruction played separately |
| RTKCombinePlay | 27 | Real-time kinematic lane instruction played together with other elements |

Table 3: Action type element

| Element name | Onehot index | Description |
|---|---|---|
| Camera | 0 | Current road camera |
| NextCamera | 1 | Next road camera |
| Intervalcamera | 2 | Average speed check camera |
| IntervalCameraStart | 3 | Average speed check camera start position |
| IntervalCameraEnd | 4 | Average speed check camera end position |
| IntervalCameraOverSpeed | 5 | Overspeed warning during average speed check |
| IntervalCameraPass | 6 | Passing average speed check camera start position |
| IntervalCameraHalfway | 7 | Half pass average speed check |
| CameraPass | 8 | Passing road camera |
| Speedlimitsign | 9 | Speed limit sign |
| Buslane | 10 | Restricted bus lane reminder |
| RetrogradeRoad | 11 | Retrograde reminder |
| TurnLight | 12 | Attention for the right (left) turn signal |
| GlobalBridge | 13 | Bridge ahead |
| GlobalFacility | 14 | Sharp turn ahead |
| GlobalCity | 15 | Switching city reminder |
| GlobalCheckpoint | 16 | Checkpoint ahead |
| GlobalCarwalk | 17 | Destination is reached on foot reminder |
| GlobalForbidden | 18 | Restricted road reminder |
| GlobalAvoidfacilitynavi | 19 | Inescapable height limit ahead |
| GlobalService | 20 | Service area reminder |
| GlobalSpeedLimitSection | 21 | Speed limit section reminder |
| GlobalCurve | 22 | Curve ahead |
| GlobalSpeedLimitSign | 23 | Speed limit sign |
| MixforkRemind | 24 | Mix fork reminder |

Table 4: Info type element

| Onehot index | Play text |
|---|---|
| 0 | Null |
| 1 | Turn left |
| 2 | Turn right |
| 3 | Turn left ahead |
| 4 | Turn right ahead |
| 5 | Turn left and back |
| 6 | Turn right and back |
| 7 | Turn around |
| 8 | Go straight |
| 9 | Keep left |
| 10 | Keep right |
| 11 | Exit the roundabout |
| 12 | Enter the roundabout |
| 13 | Slow down |
| 14 | Merge into straight |
| 15 | Tunnel |
| 16 | Waypoint |
| 17 | Fork |
| 18 | Destination |

Table 5: Sub-type of main action

## A.2 CLOUD-EDGE COLLABORATION

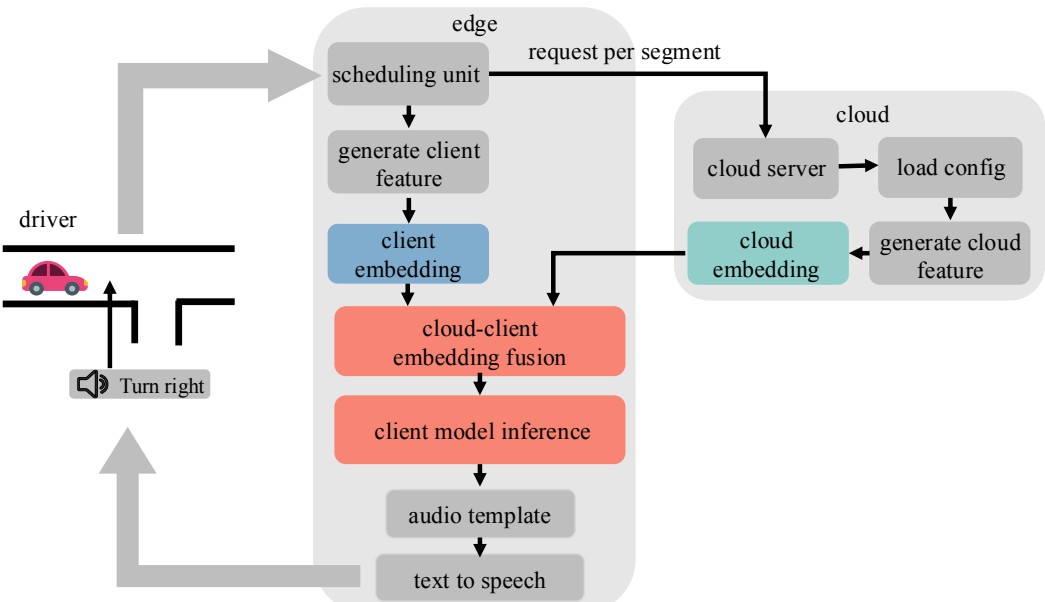

Figure 4: **Cloud-edge collaboration framework.** The left side shows the user's driving behavior, and the edge device in the middle determines whether it needs to generate audio instruction based on the driving progress, as well as the start of each segment requesting cloud feature embedding from the cloud server on the right side.

As shown in Figure 4, we implement a cloud-edge collaborative architecture that cloud-edge collaboration framework for practical applications. This design leverages the strengths of both platforms to enhance real-time performance, system scalability, and driving experience.

By deploying the model inference on edge devices, we capitalize on the computational capabilities of user hardware. Processing data locally allows for real-time responsiveness, which is crucial for delivering timely audio instructions in navigation. It minimizes latency and ensures that drivers receive immediate feedback, enabling them to make quick and safe decisions on the road.

Offloading inference tasks to the edge also reduces the computational burden on the cloud servers. This not only decreases operational costs but also enhances the system's scalability by allowing it to support a larger user base without proportionally increasing cloud resources. It prevents the wastage of cloud server resources that would occur if all computations were centralized, especially considering that edge devices often have underutilized processing power.

While the edge devices handle real-time inference, the cloud server performs pre-processing tasks and generates embeddings for static and complex features such as road graph data, a part of element features, and personalized driver features. These computations benefit from the cloud's superior processing power and centralized data storage, which allows for up-to-date and comprehensive feature embeddings that can be periodically updated without impacting the edge devices. The scheduling unit on the edge is responsible for requesting the cloud-side model embedding for this segment from the cloud at the beginning of each segment and for orchestrating the edge-side play model inference.

An important advantage of this cloud-edge collaboration is the flexibility it provides in updating the model. With the inference model on the edge and feature embeddings on the cloud, we can update components independently. This modularity accelerates the iteration cycle of the model, reducing it from a monthly to a weekly timeframe. As a result, we can deploy updates and improvements more rapidly, responding promptly to user feedback and evolving requirements. This agility opens up greater possibilities for supporting additional features in the future, enhancing the system's adaptability and longevity.

Moreover, differentiating the tasks based on their timing requirements optimizes system performance. Real-time processing is handled by the edge, meeting the immediate demands of navigation instructions. In contrast, the cloud handles tasks that can be pre-computed, like embedding updates, which do not require instant processing. This separation ensures efficiency by aligning computational tasks with the most suitable platform.

In conclusion, our cloud-edge collaborative approach effectively balances efficiency, effectiveness, and cost. By leveraging the computational strengths of edge devices for real-time inference and the cloud for intensive pre-processing tasks, we optimize resource utilization. The flexibility in updating the model enhances iterative efficiency, allowing for faster deployment of improvements and new features. This architecture not only improves the scalability and performance of the TBT navigation system but also significantly enhances the driver's experience by providing timely, accurate, and context-aware audio instructions.

### A.3   POSITION EMBEDDING

We design the position embedding as in Equation 7, where features with smaller distances to the next navigation point $d_{np}$ will receive more attention during the multi-head attention computation. This is essentially an additional inductive bias that we provide to the multi-head attention computation based on domain knowledge. Similar design has been verified as valid in past research on transformers Yang et al. (2022).

$$\gamma = \begin{cases} \frac{1}{2^{\lfloor \frac{d_{np}}{50} - 1 \rfloor}}, & 0 \leq d_{navi} \leq 300 \\ \frac{1}{2^{\lfloor \frac{d_{np}}{100} + 2 \rfloor}}, & 300 \leq d_{navi} \leq 600 \\ \frac{1}{2^{\lfloor \frac{d_{np}}{200} + 6 \rfloor}}, & 600 \leq d_{navi} \leq 1000 \\ \frac{1}{2^{\lfloor \frac{d_{np}}{500} + 9 \rfloor}}, & 1000 \leq d_{navi} \leq 3000 \\ \frac{1}{2^{14}}, & d_{np} \geq 3000 \end{cases} \tag{7}$$

### A.4   DATA COLLECTION

Figure 5 illustrates the procedural workflow for constructing the dataset utilized in model training. The production of samples involves approximately four steps:

The first step entails extracting raw navigation trajectory logs that meet specific selection criteria into a temporary table. This step solely focuses on data preservation without processing, ensuring that any issues encountered in subsequent processes can be swiftly traced back to either this step or the upstream processes. Additionally, this facilitates

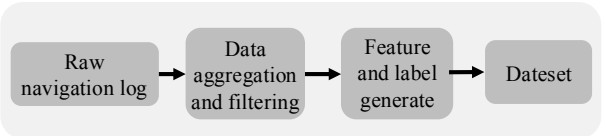

Figure 5: **Dataset production process.** The TBT log is restored offline based on the actual navigation trajectory logs of real online drivers. Then model input features, and labels are generated accordingly. Finally, the dataset is completed.

data validation against the raw data. The selection criteria include choosing navigation paths initiated by cars, excluding paths where the driver's terminal was muted, eliminating data with anomalies, and discarding paths with yaws.

In the second step, the raw data undergoes aggregation and filtration: Initially, point data are merged based on whether they belong to the same navigation path. Subsequently, segment data is consolidated into point data, simplifying the subsequent processing of point data by avoiding multiple associations with segment information. During this process, data is subjected to rounds of cleaning to remove outliers. For data exhibiting poor performance in real navigation (such as drift GPS points or yaws), negative labeling is applied. Following this, simulated driving behavior and GPS signals are recreated offline based on the parsed trajectory points and segment information, thereby restoring the complete trajectory points and other information on the path. Finally, the model prediction points and corresponding audios to be predicted are determined.

The third step involves generating features and labels for the model prediction points, which are then merged to form the samples.

The last step consolidates data produced in different batches, partitioning it into training, validation, and test datasets. Feature standardization is performed using the mean and standard deviation of the features. Subsequently, single positioning point sample data is concatenated into sequential sample data. And finally, the dataset for model training is complete.

The complete dataset consists of sequence samples totaling 1.561 billion. This includes 1.101 billion in the training set, with 1.1 billion for pretrain and 10 million high-quality samples for finetune. 140 million in the validation set, and 320 million in the test set. These data were derived from 8 days of online real navigation logs collected from June 11 to June 18, 2023.

It is worth noting that the online TBT audio instruction mechanism used in data collection is not implemented through a neural network model but rather employs a Hidden Markov Model (HMM) for mapping and Viterbi inference to select the elements and timing for audio instruction Zhou (2012). We refer to this as the HMM model. The HMM model determines the number of elements to be played based on the remaining distance $d_{np}$ to the next navigation point and infers the layers accordingly. Starting from the last audio at the navigation point, the inference is conducted towards the current driver position. It then combines segment history audios, driver information, and other contextual features, scoring each candidate node in each layer based on a ranking score system to ultimately decide the audio content and timing. The observation probability of a node is calculated as the average score of the element combination at the node:

$$P(O_t \mid S_t) = \frac{1}{2} \left( S_{\text{avg}} + S_{\text{cum}} \right) \tag{8}$$

where $P(O_t \mid S_t)$ is the observation probability of the node at time $t$. $O_t$ is the observation of the current node, which is the audio instruction consisting of the selected element and timing, and $S_t$ is the state of the current node, which judges the value of the audio instruction represented by the current node. $S_{\text{avg}}$ is the average element combination score, calculated as the geometric mean of the scores of individual elements in the node's element combination. The individual element score is setted based on prior knowledge. $S_{\text{cum}}$ is the cumulative element combination score, evaluating the attachment strength of the element combined with the main action element.

The transition probability primarily evaluates the accuracy and reasonableness of transitions between nodes in different layers:

$$P(S_t \mid S_{t-1}) = S_{\text{diff}} \cdot S_{\text{rep}} \cdot S_{\text{trans}} \tag{9}$$

where $P(S_t \mid S_{t-1})$ is the state transition probability from the node at time $t-1$ to the node at time $t$. $S_{\text{diff}}$ is the information difference score, evaluating the information gain during the transition between two nodes, including information increase, decrease, and repetition. $S_{\text{rep}}$ is the segment repetition score, considering the global historical features within the segment and down-weighting elements that have already been played. $S_{\text{trans}}$ is the transition reasonableness score. Evaluating the rationality of state transfers based on priori rule constraints Finally, the optimal path and nodes are determined by computing the Viterbi algorithm over all possible paths.

### A.5 Supplementary data for real-world A/B test

The detailed element play rates presented in Table 6 further corroborate the effectiveness of our sequence model in enhancing the informativeness of audio instructions. Notably, the play rates of elements strongly correlated with yaw rate—specifically **Mainaction**, **Assistaction**, and **Lane**—have significantly improved across all modes compared to the HMM-based method. For instance, the play rate of **Mainaction** in the "Minimalist" mode increased by 88.14%, while **Assistaction** saw a substantial rise of 36.26% in the "Concise" mode. The **Lane** element also experienced improvements, with a 3.36% increase in the "Detail" mode. These enhancements indicate that our model more effectively delivers critical navigational cues, ensuring drivers receive timely and essential information necessary for safe driving.

Moreover, the average element play rate across all modes has improved, demonstrating that our sequence model successfully integrates more informative content into the audio instructions without overwhelming the driver. This balance between informativeness and cognitive load is crucial; by increasing the play rates of key elements, the model provides drivers with the necessary guidance

| Scene/Mode | Detail | | | Concise | | | Minimalist | | | Intelligent | | |
|---|---|---|---|---|---|---|---|---|---|---|---|---|
| Scene/Method | HMM | Sequence Model | Difference | HMM | Sequence Model | Difference | HMM | Sequence Model | Difference | HMM | Sequence Model | Difference |
| ET_AroundFork | 151.16% | 149.55% | -1.61% | 150.09% | 145.70% | -4.39% | 122.75% | 128.14% | 5.39% | 150.53% | 148.53% | -1.99% |
| **ET_Assistation** | **44.13%** | **70.15%** | **26.02%** | **28.55%** | **64.81%** | **36.26%** | **16.89%** | **45.68%** | **28.78%** | **41.03%** | **68.45%** | **27.43%** |
| ET_Buslane | 4.41% | 4.22% | -0.19% | 4.63% | 4.48% | -0.15% | 0.15% | 0.14% | -0.01% | 4.19% | 4.07% | -0.12% |
| ET_Camera | 174.98% | 163.40% | -11.58% | 82.03% | 84.53% | 2.51% | 28.07% | 29.24% | 1.17% | 145.37% | 135.73% | -9.64% |
| ET_CameraPass | 37.01% | 33.57% | -3.44% | 37.54% | 35.99% | -1.55% | 0.68% | 0.36% | -0.32% | 32.21% | 30.17% | -2.03% |
| ET_ExitRoad | 23.54% | 25.50% | 1.97% | 0.09% | 0.03% | -0.05% | 0.11% | 0.09% | -0.01% | 19.23% | 20.81% | 1.58% |
| ET_GlobalBridge | 5.42% | 5.48% | 0.06% | 0.17% | 0.13% | -0.04% | 0.24% | 7.99% | 7.75% | 4.45% | 4.47% | 0.01% |
| ET_GlobalCarwalk | 0.08% | 0.07% | -0.01% | 0.08% | 0.07% | -0.01% | 0.07% | 0.08% | 0.01% | 0.08% | 0.07% | -0.01% |
| ET_GlobalCity | 4.76% | 4.54% | -0.22% | 0.10% | 0.08% | -0.02% | 0.28% | 0.23% | -0.05% | 3.88% | 3.75% | -0.13% |
| ET_GlobalCurve | 0.91% | 1.23% | 0.31% | 4.39% | 2.16% | -2.23% | - | 0.03% | - | 0.79% | 1.08% | 0.29% |
| ET_GlobalFacility | 55.68% | 56.41% | 0.73% | 0.17% | 0.14% | -0.04% | 0.45% | 0.40% | -0.05% | 40.63% | 40.79% | 0.16% |
| ET_GlobalForbidden | 0.02% | 0.01% | -0.01% | 0.02% | 0.01% | -0.01% | - | - | - | 0.01% | 0.00% | -0.01% |
| ET_GlobalService | 19.81% | 19.69% | -0.12% | 7.96% | 7.54% | -0.42% | 0.30% | 0.23% | -0.07% | 16.76% | 16.56% | -0.20% |
| ET_GlobalSpeedLimitSign | 42.28% | 45.86% | 3.57% | 0.23% | 0.14% | -0.09% | 0.27% | 0.20% | -0.07% | 33.77% | 38.42% | 4.65% |
| ET_IntervalCameraEnd | 5.89% | 5.90% | 0.01% | 5.48% | 5.46% | -0.02% | 0.10% | 0.18% | 0.08% | 5.32% | 5.32% | 0.00% |
| ET_IntervalCameraHalfway | 3.55% | 3.49% | -0.06% | 2.98% | 2.91% | -0.07% | 0.06% | 0.28% | 0.23% | 3.26% | 3.26% | 0.00% |
| ET_IntervalCameraOverSpeed | 3.46% | 3.87% | 0.40% | 2.99% | 3.49% | 0.50% | 2.79% | 4.17% | 1.38% | 3.25% | 3.50% | 0.25% |
| ET_IntervalCameraPass | 3.23% | 3.15% | -0.08% | 2.93% | 2.86% | -0.08% | 0.08% | 0.19% | 0.12% | 2.94% | 2.87% | -0.07% |
| ET_IntervalCameraStart | 7.15% | 6.79% | -0.36% | 6.38% | 6.32% | -0.06% | 0.14% | 0.21% | 0.07% | 6.19% | 5.94% | -0.26% |
| **ET_Lane** | **89.89%** | **93.25%** | **3.36%** | **0.70%** | **0.78%** | **0.07%** | **0.40%** | **0.77%** | **0.37%** | **78.83%** | **82.13%** | **3.30%** |
| ET_Linkturn | 2.75% | 0.84% | -1.91% | 2.35% | 0.44% | -1.92% | 1.66% | 0.21% | -1.45% | 2.54% | 0.64% | -1.90% |
| ET_Longsolidlane | 6.54% | 9.71% | 3.17% | 5.42% | 5.88% | 0.46% | 5.99% | 6.35% | 0.36% | 5.55% | 8.80% | 3.25% |
| **ET_Mainaction** | **409.73%** | **428.93%** | **19.19%** | **305.28%** | **332.57%** | **27.29%** | **207.80%** | **295.93%** | **88.14%** | **396.28%** | **426.98%** | **30.70%** |
| ET_Mixfork | 110.80% | 87.31% | -23.50% | 80.14% | 85.34% | 5.21% | 70.66% | 75.14% | 4.48% | 92.75% | 113.13% | 20.38% |
| ET_MixforkRemind | 8.15% | 8.59% | 0.44% | 15.00% | 0.96% | -14.04% | 100.00% | 0.69% | -99.31% | 7.28% | 8.02% | 0.74% |
| ET_MixLane | 4.21% | 5.05% | 0.84% | 16.67% | 14.29% | -2.38% | | | | 3.54% | 4.27% | 0.73% |
| ET_NextAssistation | 1.10% | 1.55% | 0.45% | 1.12% | 1.52% | 0.41% | 2.88% | 2.67% | -0.20% | 1.05% | 1.57% | 0.52% |
| ET_Nextbrname | 85.94% | 87.41% | 1.48% | 28.20% | 32.63% | 4.43% | 0.36% | 0.34% | -0.01% | 74.98% | 78.75% | 3.77% |
| ET_NextExitRoad | 0.16% | 0.18% | 0.02% | | | | | | | 0.12% | 0.13% | 0.01% |
| ET_NextLane | 5.81% | 6.58% | 0.78% | 7.43% | 7.42% | -0.02% | 0.16% | 0.63% | 0.47% | 5.73% | 6.43% | 0.69% |
| ET_NextMainaction | 30.21% | 24.36% | -5.85% | 21.93% | 22.62% | 0.69% | 16.56% | 17.74% | 1.19% | 29.74% | 25.32% | -4.42% |
| ET_NextMixfork0 | 0.20% | 0.24% | 0.04% | 0.21% | 0.25% | 0.05% | 0.01% | 20.00% | 19.99% | 0.18% | 0.23% | 0.05% |
| ET_NextNextAct | 0.28% | 0.24% | -0.04% | | | 0.11% | 0.11% | | | 0.22% | 0.22% | 0.00% |
| ET_NextSegNextbrname | 7.83% | 7.48% | -0.35% | 2.28% | 1.83% | -0.46% | 3.15% | 5.53% | 2.37% | 7.28% | 6.79% | -0.49% |
| ET_NextSlope | 2.83% | 1.81% | -1.02% | 8.47% | 16.00% | 7.53% | 20.00% | | | 2.27% | 1.37% | -0.90% |
| ET_Nonnavigation | 148.75% | 144.01% | -4.74% | 28.06% | 28.15% | 0.09% | 1.29% | 1.32% | 0.03% | 128.39% | 124.84% | -3.55% |
| ET_RetrogradeRoad | 0.23% | 0.21% | -0.02% | 0.10% | 0.11% | 0.01% | 0.08% | 0.14% | 0.05% | 0.22% | 0.19% | -0.03% |
| ET_ShortNonNaviLane | 3.79% | 4.58% | 0.79% | 0.07% | 0.08% | 0.01% | 0.11% | 0.13% | 0.02% | 3.35% | 3.71% | 0.36% |
| ET_Slope | 21.23% | 23.63% | 2.40% | 10.18% | 22.03% | 11.85% | 13.61% | 0.35% | -13.25% | 18.79% | 22.17% | 3.37% |
| ET_SolidLane | 0.05% | 0.05% | 0.01% | 0.05% | 0.07% | 0.01% | 0.07% | 0.08% | 0.01% | 0.04% | 0.04% | 0.00% |
| ET_TunnelLane | 17.47% | 17.20% | -0.26% | 10.33% | 14.19% | 3.86% | 38.46% | 6.49% | -31.98% | 16.38% | 16.80% | 0.42% |
| ET_TurnLight | 1.15% | 1.08% | -0.07% | 0.98% | 1.05% | 0.07% | 3.79% | 3.76% | -0.03% | 1.20% | 1.17% | -0.03% |
| ET_UnSlope | | | | | | | | | | 0.00% | | |
| Total | 46.26% | 49.38% | 3.12% | 28.86% | 33.46% | 4.60% | 33.49% | 47.70% | 14.21% | 40.87% | 44.54% | 3.67% |

Table 6: Element play rates

to navigate complex intersections and road networks confidently. When combined with the results shown in Figure 3,(b), which illustrates that the words per segment have only marginally increased, it becomes evident that our model enhances informational content efficiently. This efficient delivery contributes to reduced yaw rates, as drivers are better prepared and less likely to deviate from the intended route, ultimately validating the benefits of our approach in real-world driving scenarios.

### A.6 SUPPLEMENTARY DATA FOR ABLATION STUDY

| Sequence length | Trigger 10m | Trigger 30m | Action | Info | VoiceOrder |
|---|---|---|---|---|---|
| 3 (our model) | 83.3% | 96.3% | 97.0% | 98.6% | 90.7% |
| 1 | −5.5% | −1.4% | −0.3% | −0.2% | −1.5% |
| 2 | −5.0% | −2.0% | −0.1% | 0.0% | −0.3% |
| 4 | 0.1% | 0.0% | 0.1% | −0.1% | −0.0% |
| 5 | 0.0% | 0.0% | 0.1% | 0.0% | 0.0% |

Table 7: Sequence length ablation study

We conducted an ablation study to determine the optimal sequence length for our model by varying it from 1 to 5 and observing the impact on performance metrics. As presented in Table 7, reducing the sequence length to 1 and 2 resulted in a significant decline in performance. This indicates that shorter sequences fail to capture sufficient temporal dependencies, adversely affecting the model's ability to predict audio instruction timing within a critical 10-meter range. The minimal decreases in other metrics further underscore the importance of sequence data in accurately modeling spatiotemporal patterns essential for effective navigation instructions.

Conversely, increasing the sequence length beyond 4 yielded diminishing returns. Extending the sequence length to 5 did not provide additional benefits. These observations suggest that while incorporating historical data enhances the model's predictive capabilities, excessive sequence lengths introduce redundant information without meaningful gains. Therefore, a sequence length of 3 strikes an optimal balance between capturing adequate historical context and maintaining computational efficiency. This choice allows the model to effectively leverage spatiotemporal dependencies inherent in driving scenarios, enhancing the accuracy and contextual relevance of real-time audio instructions without incurring unnecessary computational overhead.

| MoE expert number | Trigger 10m | Trigger 30m | Action | Info | VoiceOrder |
|---|---|---|---|---|---|
| 3 (our model) | 83.3% | 96.3% | 97.0% | 98.6% | 90.7% |
| 2 | −0.2% | −0.1% | −0.1% | 0.0% | −0.2% |
| 1 | −0.5% | −0.2% | −0.2% | 0.1% | −0.4% |

Table 8: MoE expert number ablation study

### A.7 COMPLEX DRIVING SCENARIOS

The 6 complex scenarios mentioned in Experiment 4.4 are described in this section. The meanings of all the symbols in the following figure are the same as those presented in the legend of Figure 1, with the black solid line representing the edge of the road, the red car representing the current GPS-located driver's position, the directed path made up of blue dashed arrows representing the navigation path planned by the navigation system, one blue dashed arrow representing a segment, and the orange dots representing the navigation points.

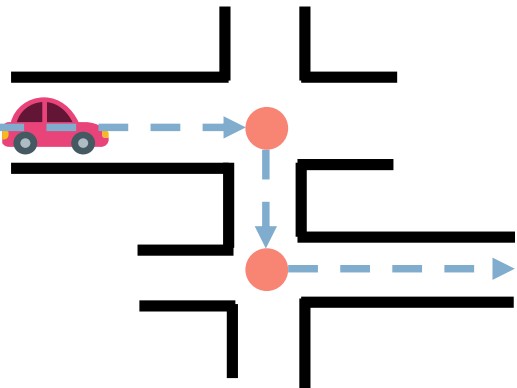

Figure 6: **Near double bend.** The next segment is short, and the driver is about to face two consecutive turns. The elements of the next segment should be pre-played into the current segment to avoid incomplete transmission due to short navigation.

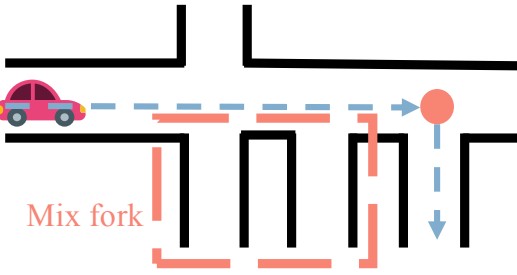

Figure 7: **Mix fork.** If one or more forks are closer to the next navigation point in the same segment, and the fork roads are facing in the same direction as the turn needed for the next navigation point, these forks are called mix forks. They are shown in the orange dotted box. Drivers should be warned not to turn early at this point.

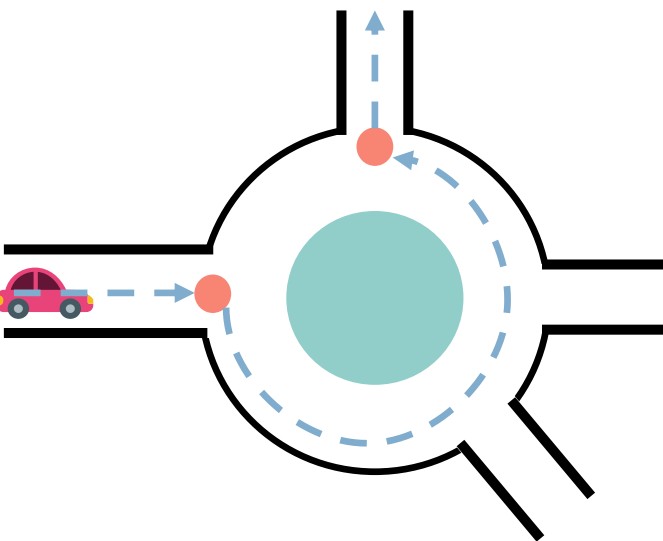

Figure 8: **Roundabout.** A roundabout comprises a circular roadway and a central island, designed so that traffic approaching from any direction must enter the roundabout and circulate in a single, consistent direction around the central island until reaching the desired exit. Multiple exits may exist within the roundabout. The audio instruction policy should alert drivers to exit at the correct fork.

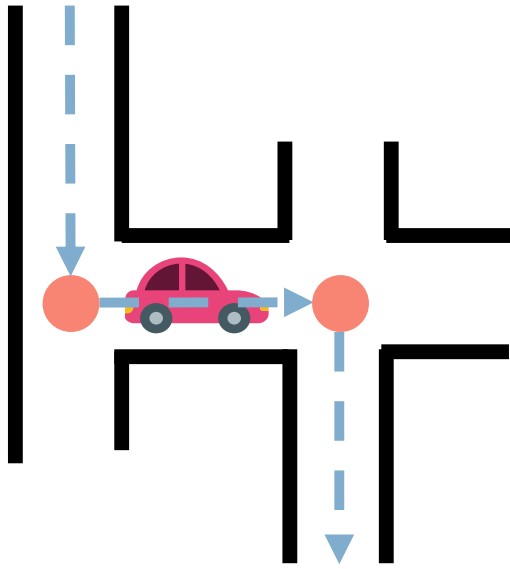

Figure 9: **Short segment.** Short segment scenarios refer to instances where the driver enters a short segment. Given the brevity of these segments, it is essential to condense the audio to prevent them from being overshadowed by previous instructions or delayed in delivery.

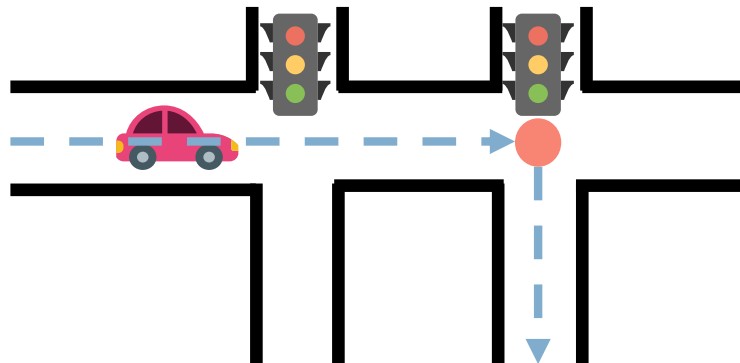

Figure 10: **Double traffic light.** The double traffic light scenario involves the presence of two traffic lights, where the driver should execute a turning operation at the second light. It is crucial to remind the driver of the traffic light sequence and provide clear guidance. The audio instruction policy should recall the turning information at the second traffic light when approaching the first.

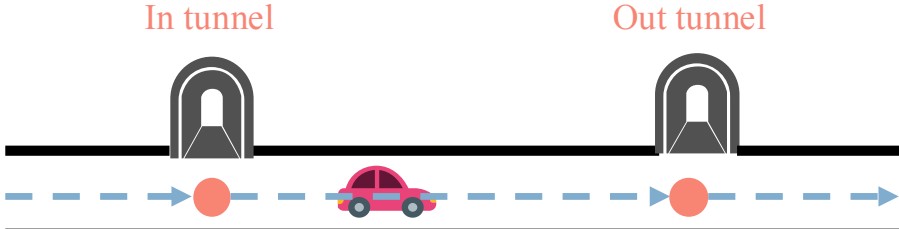

Figure 11: **Tunnel.** The tunnel scenario pertains to the situation where a driver is traversing through a tunnel. Given that entering a tunnel often results in the loss of signal and GPS positioning on the user's navigation device, it is hard to communicate with the navigation system in real-time. Therefore, it is necessary to preemptively retrieve the required cloud-based information before entering the tunnel. Additionally, audio instructions for post-tunnel driving operations should be provided within the tunnel to ensure the driver has ample time to change lanes upon exiting.

## A.8 MODEL HYPERPARAMETERS AND FEATURES

| Meaning | Embedding device | Dimension |
| --- | --- | --- |
| Road network feature | cloud | 414 |
| Action element feature | cloud | 168 |
| Info element feature | cloud | 150 |
| Personalized feature | cloud | 99 |
| Position feature | edge | 299 |
| Action element feature | edge | 196 |
| Info element feature | edge | 175 |
| History play feature | edge | 638 |

Table 9: Model input feature

| Name | Value | Description |
| --- | --- | --- |
| max_len | 3 | Input feature length |
| predict_max_len | 3 | Model predict length |
| speed_max_len | 10 | The max speed feature length |
| mlp_hid_dim | 156 | MLP hidden dimension |
| lr | 1e-3 | Learning rate |
| adam_weight_decay | 1e-7 | Adam weight decay |
| adam_beta1 | 0.9 | Adam $\beta_1$ |
| adam_beta2 | 0.999 | Adam $\beta_2$ |
| hidden | 256 | Hidden dimension for GPT decoder |
| layers | 1 | Layers for GPT decoder |
| attn_heads | 4 | GPT decoder attention head |
| batch_size | 800 | Batch size for training |
| att_head | 32 | MHA attention head |
| att_hid | 128 | MHA hidden dimension |
| att_emb | 79 | MHA input embedding dimension |
| speed_att_head | 2 | Speed MHA attention head |
| speed_att_hid | 2 | Speed MHA hidden dimension |
| mlp_layer_num | 2 | Number of MLP layer |
| decoder_mlp_hid_dim | 64 | MLP dimension for model output decoder |
| decoder_mlp_layer_num | 2 | Number of layer for model output decoder |
| mlp_emb_dim | 8 | Embedding dimension for input feature |

Table 10: Hyperparameters

