# OpenReview forum: "Turn-by-Turn Driving Navigation: Leveraging Sequence Model for Real-time Audio Instructions"
_ICLR.cc/2025/Conference — ICLR 2025 Conference Withdrawn Submission_

### Official Review · Reviewer_N9PL · 2024-10-28

**Soundness:** 2
**Presentation:** 2
**Contribution:** 2
**Rating:** 3
**Confidence:** 3

**Summary:**

This paper presents a deep-learning based method for generating audio navigation instructions for drivers. The proposed modelA neural is constructed based on the Transformer architecture and a mixture of experts (MOEs), along with a multi-objective loss function for training. Experimental results indicate that, compared to HMM-based methods, the proposed approach achieves higher subjective scores.

**Strengths:**

1. The paper presents a novel application of deep learning.
2. The paper is well-strutured and easy to read.

**Weaknesses:**

1. Lack of novelty. There are few innovative designs observed in the network architecture or training.
2. The experimental section lacks a broader comparison. Audio navigation instructions are a common feature in mapping applications, and there are likely many established methods available. The paper only compares with one HMM-based method, which was proposed in 2012 and is not novel.
3. The ablation study does not provide valuable insights. The experimental results indicate that removing most modules solely (e.g. the MOE module) from the network does not lead to significant performance degradation, which raises concerns about potential redundancy in the network design.

**Questions:**

1. Is there a specific advantage to using geometric averaging of different loss functions in this task? Was there any comparison made with arithmetic averaging?
2. The paper mentions that the HMM-based instruction policy was used during the data collection phase. Does this imply that the supervision signal for training the model comes from this algorithm?

---

### Official Review · Reviewer_C5PV · 2024-11-01

**Soundness:** 2
**Presentation:** 2
**Contribution:** 1
**Rating:** 3
**Confidence:** 5

**Summary:**

In this paper, the authors propose a method to optimize turn-by-turn navigation instructions for drivers using a deep learning approach. They tested their method on a custom-created dataset and analyzed results through real-world A/B testing. The authors claim to be the first to investigate this problem and report making significant progress in this area.

**Strengths:**

Spent lots of resources in developing this method, yet have to see the benefit of that.

**Weaknesses:**

The quality of the paper is quite poor. The text is lengthy yet fails to convey the main message effectively. Key terms, such as "yaw rate" and "seesaw effect," are either undefined or introduced too late in the paper. The related work section is missing, and relevant literature is not cited. The final paragraph of the background offers no new information and merely summarizes the introduction.

The methodology is difficult to understand, lacking motivation for using such a complicated framework and failing to clarify what advantages it provides. Important details are relegated to the appendix rather than included in the main text. The paper is mostly written in the passive voice, with vague statements like, “To address the challenges in generating real-time, context-aware instructions, we model the audio instruction in TBT driving navigation as a multi-task learning problem. Enables the model to optimize the necessary components for generating the audio.” They repeatedly use the term "context-aware" without explaining what it actually means.

**Questions:**

1. What new information does the background section provide? Most of it repeats content from the introduction, and the remaining parts would be more appropriate in the methodology section.
2. The user study protocol is not clearly described. It’s unclear if 100 drivers actually drove the car following the navigation instructions, or if they merely judged the instructions by listening to them. If the evaluation was only auditory, it is inconclusive to determine the effectiveness of the proposed method.
3.  What roles do the GPT decoder and CrossNet network play in the proposed framework?
4. There are no details on data preprocessing or how the embedding features are extracted.
5.  The paper lacks details on the layer-by-layer structure of the framework and how data is processed through each stage.
6.  How is accuracy calculated in the ablation study?
7. what are the statistics of user study?

---

### Official Review · Reviewer_JAUk · 2024-11-02

**Soundness:** 1
**Presentation:** 1
**Contribution:** 1
**Rating:** 3
**Confidence:** 3

**Summary:**

This paper presents a deep learning model for turn-by-turn navigation, enhancing traditional audio guidance systems by making instructions more adaptive and context-aware. Using a sequence-based approach and a cloud-edge setup, it delivers real-time, precise directions, reducing navigation errors and easing driver cognitive load.

**Strengths:**

This paper introduces a deep learning model for turn-by-turn navigation, offering two key benefits: more adaptive, context-aware audio guidance and reduced navigation errors. The model’s sequence-based design and cloud-edge setup ensure real-time, precise directions, greatly enhancing driver support​.

**Weaknesses:**

Please refer to Questions section.

**Questions:**

1. The navigation capabilities targeted by this research are effectively addressed by existing navigation apps, which can already provide lane-level guidance with real-time audio instructions. This research does not clearly establish a novel problem or significant gap in current technology.

2. The authors assert that theirs is the first real-world application of deep learning for audio navigation. However, similar problems have been thoroughly researched and resolved in the NLP field, with deep learning applications already prevalent. Thus, the claimed contributions seem overstated.

3. The summary of contributions lacks specificity, offering mostly general points without a clear overview of the work. This makes it difficult for readers to grasp the precise focus and innovations in the research.

4. The Related Work section references too few sources and does not include recent advancements in the field. Although the authors highlight using deep learning to solve their problem, they fail to reference relevant studies on deep learning in audio navigation, a significant oversight.

5. The "Problem Formalization" section is inadequately explained. Readers cannot clearly understand the input-output flow, and while Table 9 in the appendix offers some clarification, the choice to use intermediate features as inputs adds unnecessary complexity, making the initial inputs and outputs unclear.

6. Authors state that large language models (LLMs) are unsuitable for TBT audio instruction, opting instead for a transformer-based approach. However, this claim lacks sufficient rationale, given that many LLMs perform well on similar tasks and are widely used in both academia and industry. The authors neither justify nor experimentally validate why LLMs would be unsuitable for this task.

7. Proposed method appears overly generic and largely involves combining existing model components without introducing novel ideas. This approach lacks sufficient originality to merit publication at a conference like ICLR.

8. It is unclear how the authors have knowledge of GPT’s exact architecture, given that it is a black-box model. Furthermore, considerations such as model size, inference speed, training time, and computational cost, which are crucial for real-time applications, are not discussed.

9. Important components in the methods section, such as Deep CrossNet and GPT Decoder, are not adequately described. This lack of detail leaves readers uncertain about how these components function within the model.

10. The experiments are disorganized and limited in scope. There is a lack of strong baselines and comparisons to recent, relevant work, making it difficult to ascertain whether the method achieves SOTA performance. The experiments also lack ablation studies, visualizations, and key information.

11. During driving, overly detailed instructions may be distracting, as drivers may not want or need continuous audio prompts. This issue of instruction density is not addressed.

12. Minor language errors persist, such as a missing space between "Figures 3(b)" and "3(c)" on line 422, which reflects a lack of careful proofreading.

13. The paper lacks novelty, as indicated by the outdated citations and few references to recent research, which suggests that the work does not align with the current cutting-edge.

Overall, this paper is structured more like a technical report than a research paper. Given its organization and limited scientific contribution, it does not yet meet the standard for acceptance at a conference.

---

### Note · Authors · 2024-12-06

I have read and agree with the venue's withdrawal policy on behalf of myself and my co-authors.